# Comparative Properties of Helical and Linear Amphipathicity of Peptides Composed of Arginine, Tryptophan, and Valine

**DOI:** 10.3390/antibiotics13100954

**Published:** 2024-10-11

**Authors:** Jessie Klousnitzer, Wenyu Xiang, Vania M. Polynice, Berthony Deslouches

**Affiliations:** Department of Environmental and Occupational Health, School of Public Health, University of Pittsburgh, Pittsburgh, PA 15261, USA; jek179@pitt.edu (J.K.); wex8@pitt.edu (W.X.); vmp33@pitt.edu (V.M.P.)

**Keywords:** antimicrobial peptides, antibiotic resistance, peptide antibiotics, multidrug resistance, ESKAPE pathogens, antimicrobial agents, cationic amphipathic peptides, linear amphipathicity, helical amphipathicity, cationic polymers, engineered AMPs

## Abstract

Background: The persistence of antibiotic resistance has incited a strong interest in the discovery of agents with novel antimicrobial mechanisms. The direct killing of multidrug-resistant bacteria by cationic antimicrobial peptides (AMPs) underscores their importance in the fight against infections associated with antibiotic resistance. Despite a vast body of AMP literature demonstrating a plurality in structural classes, AMP engineering has been largely skewed toward peptides with idealized amphipathic helices (H-amphipathic). In contrast to helical amphipathicity, we designed a series of peptides that display the amphipathic motifs in the primary structure. We previously developed a rational framework for designing AMP libraries of H-amphipathic peptides consisting of Arg, Trp, and Val (H-RWV, with a confirmed helicity up to 88% in the presence of membrane lipids) tested against the most common MDR organisms. Methods: In this study, we re-engineered one of the series of the H-RWV peptides (8, 10, 12, 14, and 16 residues in length) to display the amphipathicity in the primary structure by side-by-side (linear) alignment of the cationic and hydrophobic residues into the 2 separate linear amphipathic (L-amphipathic) motifs. We compared the 2 series of peptides for antibacterial activity, red blood cell (RBC) lysis, killing and membrane-perturbation properties. Results: The L-RWV peptides achieved the highest antibacterial activity at a minimum length of 12 residues (L-RWV12, minimum optimal length or MOL) with the lowest mean MIC of 3–4 µM, whereas the MOL for the H-RWV series was reached at 16 residues (H-RWV16). Overall, H-RWV16 displayed the lowest mean MIC at 2 µM but higher levels of RBC lysis (25–30%), while the L-RWV series displayed minor RBC lytic effects at the test concentrations. Interestingly, when the *S. aureus* strain SA719 was chosen because of its susceptibility to most of the peptides, none of the L-RWV peptides demonstrated a high level of membrane perturbation determined by propidium iodide incorporation measured by flow cytometry, with <50% PI incorporation for the L-RWV peptides. By contrast, most H-RWV peptides displayed almost up to 100% PI incorporation. The results suggest that membrane perturbation is not the primary killing mechanism of the L-amphipathic RWV peptides, in contrast to the H-RWV peptides. Conclusions: Taken together, the data indicate that both types of amphipathicity may provide different ideal pharmacological properties that deserve further investigation.

## 1. Introduction

Throughout the 21st and for much of the 20th century, the development of antibiotics has been critical to the treatment of bacterial diseases. However, the rise of antibiotic resistance (AR) has turned many previously curable infections into significant challenges [1,2,3]. In the United States alone, over 35,000 people die each year from these infections, with the global toll exceeding 4 million deaths annually [4,5,6,7]. AR-related infections also drive-up healthcare costs and threaten advances in medicine and agriculture. Research on antimicrobial peptides (AMPs; e.g., the defensins, the cathelicidins, the protegrins) has revealed the importance of cationic amphipathic peptides as a promising therapeutic option [8,9,10,11,12]. Naturally occurring AMPs are ubiquitous agents with a rare ability to both kill multidrug-resistant (MDR) bacteria and even inhibit endotoxin-induced inflammation [13,14,15]. AMPs that are ribosomally synthesized in animal species, referred to as AMPs with classical amphipathic structures, or simply classical AMPs, represent a crucial component of innate immunity; they are able to selectively target negatively charged lipids on bacterial surfaces through electrostatic interactions while they require relatively higher concentrations to similarly interact with eukaryotic cell membranes [16,17,18]. Thus, AMPs can either disrupt bacterial cell membranes or, less commonly, target vital intracellular structures like DNA and RNA, leading to cell death [19,20,21,22,23,24]. Unlike small-molecule antibiotics, AMPs act quickly and typically do not require bacterial growth for antibacterial activity, reducing the chance for resistance development [25]. Bacterial surface lipids are essential for membrane integrity, which may result in a slower rate of development of drug resistance during treatment with AMPs compared to small-molecule antibiotics targeting a biosynthetic pathway [5,26,27,28]. Depending on the primary sequence, AMPs may display anti-biofilm properties and combat a range of pathogens, including bacteria, fungi, viruses, and parasites [29]. Cyclic AMPs, like the polymyxins and daptomycin, which are not ribosomally synthesized like the classical AMPs, are used clinically to treat bacterial infections [30,31,32,33,34,35,36,37,38]. However, their toxicity to the host (e.g., nephrotoxicity and hepatotoxicity) has precluded their widespread use [39,40,41,42]. Thus, there have been intensive efforts to engineer and develop classical AMPs like those occurring in mammals (e.g., LL37, the defensins) for clinical applications [8,43,44,45,46,47]. These classical AMPs are an intriguing class of molecules with multiple functions and therefore have the potential for a wide range of applications.

In this context, we previously engineered several de novo series of cationic peptide antibiotics composed of Arg (R), Trp (W), and Val (V) (RWV), which resulted in several lead peptides with high selectivity against some of the most common MDR organisms, referred to as ESKAPE pathogens. These RWV peptides were designed to form idealized helical amphipathic peptides, which will be referred to in this report as H-amphipathic or H-RWV peptides. Notably, at the minimum optimum length of 16 residues or MOL, the peptide (H-RWV16) displayed up to 88% helicity in lamellar vesicles consisting of negatively charged phospholipids [48,49]. Importantly, the helicity depends on the environment as the peptide tends to form random coil in aqueous solutions. We define MOL relatively to the other peptides of the same series and not necessarily to a referenced value, although it can be compared to other MOLs of different series of peptides. By contrast, based on the number of helical turns required for potency [25,28,50], we hypothesized that the structure–function correlations of a series of idealized linear amphipathic (L-amphipathic) RWV counterparts (L-RWV) will require a shorter minimum length for antibacterial potency (MOL) than the otherwise equivalent H-RWV peptide series.

## 2. Methods

### 2.1. Bacteria, Reagents, and Antibacterial Assays

We purchased the cation-adjusted Mueller Hinton Broth (MHB2), Propidium iodide (PI) from Millipore Sigma (St Louis, MO, USA, Cat. No. 90922-500G and P4170-100MG) and fixable live/dead stain from Invitrogen (Waltham, MA, USA, Cat. No. L34962). The peptides were purchased by custom synthesis from Genscript (Piscataway, NJ, USA) which 5–10 mg/vial and dissolved by adding 500 µL/mL filter-sterilized PBS to each peptide (10 mg/mL). The small-molecule antibiotics used in the assays (Linezolid, Cefazolin, Ceftazidime, Meropenem, and Oxacillin) were all purchased from Millipore Sigma (St Louis, MO, USA, Cat. Nos. PZ0014, C5020, CDS020667, M2574, 28221, respectively). Bacterial clinical isolates were obtained from the Center for Disease Control and Prevention (CDC) with resistance profile data.

### 2.2. Growth Inhibition Assay and Assessment of Anti-Biofilm Properties

To examine antibacterial activity, we used minor modifications of a standard growth inhibition assay endorsed by the Clinical and Laboratory Standards Institute (CLSI), as previously described [49]. Bacterial inocula for testing were prepared from exponential growth phase by diluting overnight cultures at 1:100 with fresh Tryptic soy broth (TSB) (Millipore Sigma, USA, Cat. No. 22092-500G) and incubating for an additional 3–4 h. Bacteria were spun at 3000× *g* for 10 min and the pellet was resuspended in phosphate-buffered saline (PBS) (Millipore Sigma, St Louis, MO, USA, Cat. No. P4417) to determine bacterial turbidity using a Den-1B densitometer (Grant Instruments, Beaver Falls, PA, USA) at 0.5 McFarland (unit of bacterial density) corresponding to 108 CFU/mL. The prepared bacteria were incubated with each of the indicated peptides at a maximum test concentration of 16 µM and serially diluted in MHB2 and RPMI 1640 (Corning, NY, USA, Cat. No. 15-040-CM) at a 1:1 ratio. The plates were kept in an 8-drawer incubator for 18 h at 37 °C. A robotic system takes a plate from the drawer every hour and feeds a plate reader, which records the kinetics of bacterial growth at 570 nm to examine growth inhibition in real time (BioTek Instruments, Winooski, VT, USA). The minimum inhibitory concentration or MIC is the minimum peptide concentration that prevented bacterial growth, and it is objectively indicated by a flat (horizontal) line in the bacterial growth kinetic graph. Most assays are completed in duplicate. To assay for anti-biofilm activities, the 96-well plate from the growth inhibition assay is incubated for an additional 6 h (24 h total). The biomass was detected with crystal violet using a plate reader at A570, as previously described [38,51].

### 2.3. Kinetic Bacterial Killing Assay

To determine if the peptides are active in the absence of a major carbon source and the rate at which they act, the growth inhibition assay can be modified, as previously described [49,52]. Bacterial inocula is prepared as it was for the growth inhibition assay, but it is added to a non-carbon-containing source (PBS in our experiments) treated with 4 µM peptide. At different time points ranging from 30 s to 2 h, the samples are serially diluted, drip plated onto broth agar medium and incubated at 37 °C overnight. The colony forming unit/mL or CFU/mL is enumerated after incubation.

### 2.4. Assessment of Membrane Perturbation

To examine whether the selected peptides kill their bacterial target mainly by membrane permeabilization, we used propidium iodide (PI) incorporation by quantitated by flow cytometry (Novocyte, Santa Clara, CA, USA), as previously described [49,52]. Peptides were selected based low MIC, low RBC lysis, and the ability to kill bacteria in the bacterial killing assay, these factors also determined which strain of *S. aureus* (SA719) was used, SA719 was incubated in PBS at 5 × 10^8^ CFU/mL with either controls of Oxacillin at 4 µM or 50% Ethanol, while the selected peptides were tested at concentrations of 4 µM and 1 µM. Data were analyzed using GraphPad Prism software 10.

### 2.5. Determination of Red Blood Cell Lysis

Lytic effects on human red blood cells were examined as previously described [52]. Briefly, RBCs were separated by histopaque differential centrifugation using blood anonymously obtained from Vitalant (Pittsburgh, PA, USA). For RBC lysis assay, the isolated RBCs were resuspended in PBS at a concentration of 5%. The peptides were serially diluted 2-fold in 100 µL of PBS with a starting concentration of 32 µM. A volume of 100 µL of the prepared 5% RBC solution was then added for a final concentration of 2.5% RBC and incubated for 1 h. After incubation, the plates were centrifuged at 1000× *g* for 7 min and the supernatant was removed to measure its OD. In parallel, a standard curve of RBC lysis was generated by lysing the RBCs with deionized water at 2-fold RBC dilutions of 10, 20, 30, 40, 50, 60, 70, 80, 90, and 100% to determine the percentage of RBC lysis in the test samples. Both the treated test samples and standard curve were read at A570. Three independent trials of the experiment were conducted.

## 3. Results

### 3.1. Comparative Minimum Optimal Lengths (MOL) of RWV Peptides of Linear and Helical Amphipathicity

In our previous study of the H-RWV peptides, one of the series displayed the shortest MOL of 16 residues (E35 or H-RWV16, mean MIC of 2–4 µM) and displayed a high propensity to fold into a helical structure in the presence of lipid membrane mimetic solvents [48,49]. As the shorter peptide (E30, H-RWV12) of 12 residues in length had a mean MIC of ~16 µM, we initially re-engineered H-RWV12 for comparison with its idealized L-amphipathic counterpart (L-RWV12, Figure 1A). Here, “idealized” implies the amphipathicity is formed by the segregation of all hydrophobic (H) residues from all cationic (C) residues thereby establishing maximum polarity between the two motifs. The L-RWV12 peptide displays distinct C and H motifs side-by-side in the primary sequence, whereas in H-RWV12 the amphipathicity is revealed only if the peptide folds into a α-helix as shown in the helical wheel diagrams and confirmed by circular dichroism [48,49]. L-RWV12 demonstrated much higher activity (MIC, 4 µM) than H-RWV12 (MIC, 16 µM) against an *S. haemolyticus* strain that is resistant to linezolid (Figure 1B).

Next, we re-engineered the previously characterized series of H-amphipathic peptides up to the most potent H-WRV16. As shown in Table 1, the H-RWV series (formerly named E-series) display the amino acids as interspersed C and H residues in the primary sequence, as they were modeled to form idealized amphipathic helices, shown by helical wheel analysis (Table 1 and Appendix A). By contrast, in the L-RWV peptides, the C and H residues are aligned side by side as two distinct L-amphipathic C and H motifs in the primary structure. In these sequences, unlike the H-amphipathic peptides, the helical wheel conformations appear scrambled (no distinct C and H motifs) to amphipathicity, as represented in Figure 1 and Appendix A. Thus, two different types of idealized amphipathic structures are being compared, helical amphipathicity based on the arrangement of the residues in the secondary structure and linear amphipathicity based on the side-by-side or linear arrangement of the residues in the primary structure. Of note, because the idealized linear amphipathicity is formed by covalence (peptide bonds), it cannot be altered by the folding of the primary structure. Thus, regardless of the secondary structure, the C and H residues remain aligned linearly and, therefore, segregated as two distinct motifs. To compare the MOLs of these two distinct series of different types of amphipathicity, we determine the MICs against MDR clinical isolates of seven different organisms (Figure 2 and Table 2). The L-amphipathic peptides displayed the shortest MOL at 12 residues (L-RWV12) against *S. aureus*, mean MIC < 8 µM; *P. aeruginosa*, 16 µM; *A. baumannii*, *K. pneumoniae*, and *Enterobacter* spp., mean MIC of 4–6 µM; and *E. faecium*, mean MIC > 16 µM. Against *E. coli*, the MOL was achieved at 14 residues (L-RWV14) in length. As length increases beyond the MOL, MIC may increase to a level as high as 24 to 32 µM (L-RWV16). By contrast, the H-amphipathic series reached an MOL at 16 residues (H-RWV16), consistent with previous data and in decreasing order of activity, with a mean MIC as low as 2 µM for *S. aureus* and *E. coli* (also H-RWV14); 3 µM for *Enterobacter* spp., 4 µM for *K. pneumoniae*; 5 µM for *A. baumannii,* and 9 µM for *P. aeruginosa*. Thus, while the L-amphipathic RWV peptides achieve the MOL typically at 12 residues in length, the H-amphipathic RWV peptides characteristically display a lower mean MIC at the MOL of 16 residues. Next, we used a strain of *S. aureus* (SA719) that is susceptible to most of the peptides to compare the activity against bacterial biofilm using the crystal violet method for the detection of the biofilm mass. We found that the H-RWV peptides were slightly more active against biofilms, with minimum biofilm elimination concentrations (MBEC, no detectable biomass) ranging from 1 to 2 µM for H-RWV and 2 to 16 µM for L-RWV series (Table 3 and Figure 3). Notably, H-RWV16 displayed the lowest MBEC (0.5–1 µM), whereas L-RWV16 displayed the highest (~16 µm).

### 3.2. L-RWV Peptides Display Lower Lytic Effects on Human Red Blood Cells at Their Respective MOL

We compared the L- and H-RWV peptides for human RBC lysis. The L-RWV peptides displayed minor lytic effects on RBCs; at concentrations up to 32 µM, the range of RBC lysis was 1–12% (Figure 4). At the MOL (L-RWV12), the highest lytic effects were ~7%. The H-RWV peptides begin to display detectable but minor RBC lysis at 14 residues (3% lysis) and moderate lysis (21–30%) at 16 residues (H-RWV16), the MOL.

### 3.3. H-RWV Peptides Perturb Bacterial Cell Membranes More Efficiently than L-RWV Peptides

Given the differences in the primary structures of the H- and L-RWV peptides and the previously determined membrane-perturbing mechanism of the H-RWV peptides [49], we sought to determine the differences in the ability to perturb bacterial membranes. First, to establish a bactericidal mechanism, we determined the killing kinetics of the 2 series at 4 µM against SA719. Consistent with the MICs, the results revealed that H-RWV16 achieved complete killing within the first 2 min of incubation with SA719 followed by L-RWV12 at 10–12 min (Figure 5). The observation that most of the bacterial killing (90–99%) occurred during the first minute is noteworthy, although H-RWV12 and L-RWV16 did not reach complete killing during the 90 min incubation time.

To assess the membrane-perturbing properties of the 2-peptide series, we used propidium iodide (PI) incorporation detected by flow cytometry. The H-RWV peptides demonstrated much higher levels of PI incorporation than the L-RWV peptides. At corresponding lengths, the % PI incorporation at 4 µM was 95 and 64 for H-RWV12 and L-RWV12, respectively, 96 and 47 for H-RWV14 and L-RWV14, 100 and 42 for H-RWV16 and L-RWV16 (Figure 6). It is important to note that even the most active peptides of the L-RWV series, L-RWV12 and L-RWV14, only displayed half the % PI incorporation or less observed for H-RWV16. This is despite the high killing kinetics of L-RWV12, indicating that the L-RWV peptides are not primarily membrane disruptors.

## 4. Discussion

Our study underscores the differential functional properties of two types of amphipathicity of RWV peptides. We demonstrated that the two series of cationic peptides achieved their highest antibacterial activity at different peptide lengths, defined as minimum optimal lengths or MOLs. The L-amphipathic peptides reached their lowest MIC at 12 residues. However, the H-amphipathic peptides, though achieving an MOL at 16 residues, displayed a higher potency at that MOL than the activity achieved by the L-amphipathic peptides both against planktonic and biofilm growth. The first conclusion from this study is consistent with the observation of both types of amphipathicity in naturally occurring AMPs, which can fold into all types of secondary structures. This does not prevent investigators who work with helical peptides from using peptides scrambled to helicity as negative AMP controls, which are sometimes classified as non-amphipathic. Beyond the obvious conclusion of differences in sequence specificity, some of these studies ignore the fact that amphipathicity is not exclusively a helical property [53]. Such a fallacy can be illustrated by the example of L-RWV16 (low activity, low hydrophobic moment of the helical wheel conformation), a scrambled sequence corresponding to H-RWV16 (high activity, high hydrophobic moment). Yet, by comparison of the RWV16 pair, L-RWV12 (high activity, low hydrophobic moment) and H-RWV12 (low activity, high hydrophobic moment) demonstrate a reverse relationship. The typical helical amphipathicity is not displayed by the defensins, which have a propensity to fold into β-sheet structures stabilized by disulfide bridges or by the cyclic lipopeptides like the polymyxins [26,54]. Many investigators who have engineered cationic peptides often refer to the amphipathicity of AMPs as the property to fold into a helical amphipathic structure in which the C and H motifs are revealed using the helical wheel analysis and biophysical studies. That type of amphipathicity is the basis of AMPs like LL37, WLBU2, and many others [28,55]. By contrast, we define linear amphipathicity as the segregation of C and H residues into separate motifs directly formed by peptide bonds or covalence when the C or H residues are aligned side by side in the primary sequence. The role of linear amphipathicity in AMP functions has remained unclear and markedly underutilized in AMP design. Because in nature peptide structures are often not idealized into a particular structure, linear amphipathicity is often not typically apparent. Such is the case of the peptide indolicidin ILPWKWPWWPWRR with only a charge of +3—note the underlined linear hydrophobic compared to the cationic motif) [56,57]. One of the best examples can be partially illustrated by some of the amphipathic cell-penetrating peptides (CPPs; e.g., Pep-1 KETWWETWWTEWSQPKKKRKV) [58,59,60], which represent another structural class of cationic amphipathic AMPs. Just like the primary structures, the secondary structures of L-RWV12 and H-RWV16 are evidently different with a much higher helical content in the H-amphipathic peptide [48,61]. Of note, H-RWV16 displayed a higher lytic effect (3-fold) on human RBCs than L-RWV12, which could be partially explained by the shorter length of the L-amphipathic peptide, as L-RWV16 also displayed lower antibacterial activity and lower RBC lysis and suggests a lower cytotoxic tendency of the L-amphipathic peptides. Thus, despite its similarity to L-RWV12 in secondary structure (previously determined) [61], L-RWV16 demonstrated lower overall activity, which may be determined (in this case) by the length of the peptide.

We have also demonstrated that L-RWV12 kills bacteria at a slower rate than that of H-RWV16. In addition, the flow cytometry studies indicate a lower ability to perturb the bacterial membrane of *S. aureus* SA719. These observations are consistent with prior knowledge of killing mechanisms of antimicrobial peptides that do not primarily permeabilize the bacterial cell membrane [20,62,63,64,65]. For instance, the peptide indolicidin tends to penetrate its target bacterial cell and disrupt intracellular pathways that are vital to the bacterial life cycle (e.g., binding of nucleic acids) [20,62]. On the other hand, the Polymyxins, which are cyclic lipopeptides, target lipopolysaccharide and disrupt the membrane of gram-negative bacteria [36]. Whether a peptide is membrane-permeabilizing or cell-penetrating does not appear to be exclusively dependent on a particular type of secondary structure, although correlations of activity with secondary structures can be achieved [48,49,61,66].

This fundamental study represents a good starting point for further investigating the advantages and disadvantages of these two types of amphipathicity, which can both be illustrated in nature by the different structural classes of AMPs (e.g., the defensins, β-sheet, and the magainins, α-helical). As such, there are several questions that are yet to be answered as related to peptide stability, absorption, and distribution. In addition, unpublished studies of WLBU2 and H-RWV16 indicate that end-to-end cyclization may inhibit the activity of peptides with the propensity to fold into a α-helix. Such a hurdle has been overcome typically by peptide stapling [17,47,67]. We anticipate that cyclization of the L-amphipathic peptides will enhance their therapeutic properties, as demonstrated by cyclic antimicrobial peptides [32,34,36,68,69]. Thus, considering the multifunctional properties of AMPs, investigating the comparative properties of these two types of amphipathic AMPs may lead to the design of new peptides with promising therapeutic properties that could be applicable to hard-to-treat communicable and not just bacterial diseases.

## 5. Patents

This study is included in a patent application, which was filed in May 2024.

## Figures and Tables

**Figure 1 antibiotics-13-00954-f001:**
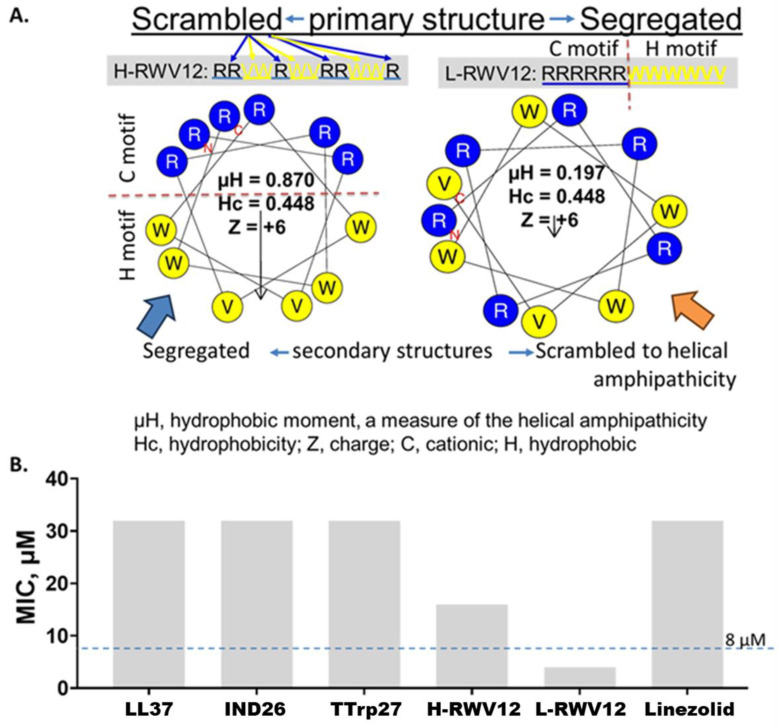
Comparative properties of H-RWV12 and L-RWV12. The primary sequence of the peptide H-WRV12 (left, (**A**)) consists of interspersed (scrambled) cationic (R, Arg) and hydrophobic (H) residues (W or V), and it is predicted to form an idealized amphipathic helix as indicated by the helical wheel diagram and high hydrophobic moment (µH, 0.87). By contrast, the primary sequence of L-RWV12, with identical amino acid composition to that of H-RWV12, displays the amphipathicity in the primary structure with C and H motifs held by peptide bonds or covalence, although its secondary structure appears scrambled to helicity with a much lower helical amphipathicity (µH, 0.197; (**A**), right helical wheel). Notably, L-RWV12 is not a negative control of H-RWV12, functionally; it is rather the more active form of the 2 RWV peptides against *S. haemolyticus* (SH 730, (**B**)). Test medium: physiologically relevant RFM (60% RPMI plus 10% fetal bovine serum or FBS and 30% MHB2); endogenous AMPs: IND26, indolicidin and Ttrp27, tritrpticin; MIC, minimum inhibitory concentration; data are representative of 2 independent trials with identical results.

**Figure 2 antibiotics-13-00954-f002:**
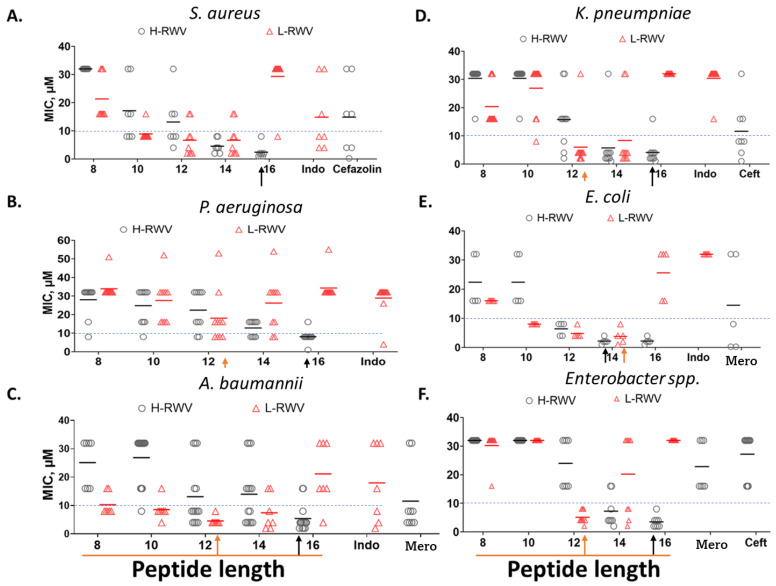
Comparative antibacterial activities of the H-RWV and L-RWV series. The 2 series were compared against MDR strains of 6 organisms as indicated. The test medium was composed of physiologically relevant RFM (60% RPMI plus 10% fetal bovine serum or FBS and 30% cation-adjusted MHB); endogenous AMP control was Indolicidin (Indo); small-molecule antibiotic controls were cefazolin, ceftazidime (Ceft), and meropenem (Mero). MIC, minimum inhibitory concentration with means represented by the black and red horizontal bars, which are used to determine the MOLs (minimum optimal lengths) indicated by the red and black arrows. (**A**–**F**) is based on the test organism as indicated. Data are representative of 2 independent trials. Corresponding Table 2 provides the mean MICs for the strains of each organism. The numbers stand for each peptide: 8, L/H-RWV8; 10, L/H-RWV10, etc.; red bar, mean MIC of L-RWV peptides and black bars, mean MIC of H-RWV peptides.

**Figure 3 antibiotics-13-00954-f003:**
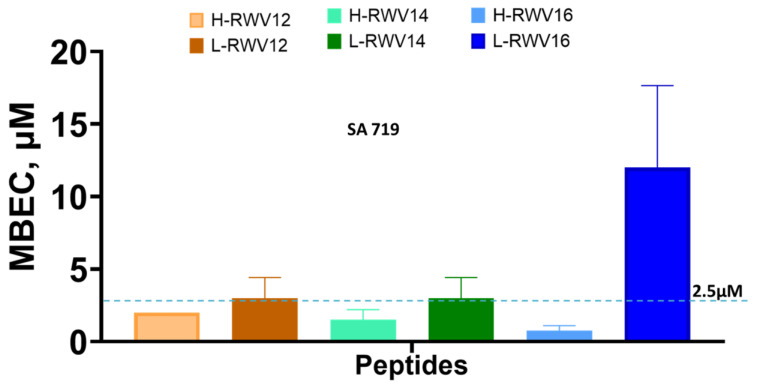
Antibiofilm activities of the R- and L-RWV peptides. MBEC or minimum biofilm elimination concentration is the minimum concentration at which no biomass was detected by the crystal violet method. Standard deviations reflect differences between 2 independent trials. The dashed line is to clarify the mid-point between 0 and 5.

**Figure 4 antibiotics-13-00954-f004:**
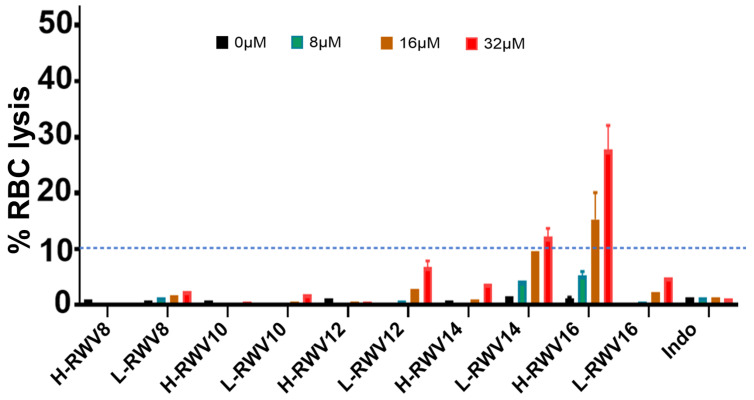
Human RBC lytic effects of the R- and L-RWV peptides. Indolicidin (Indo) was used as control; RBC lysis was assayed in PBS as described in Methods. For clarity, the dashed line indicates the 10% lysis mark across the graph as a critical point between minor (<10%) and moderate (>10%) RBC lysis.

**Figure 5 antibiotics-13-00954-f005:**
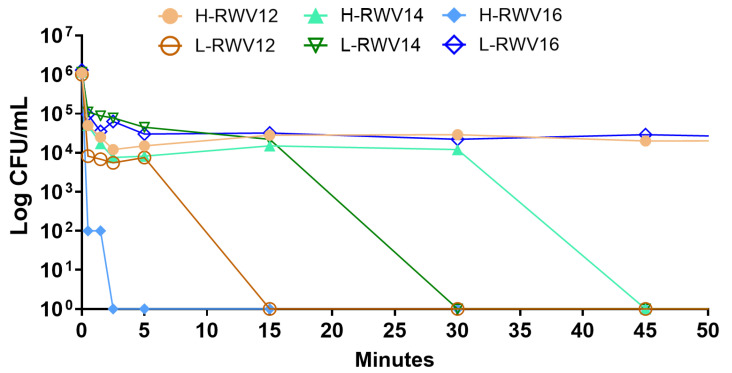
Killing kinetics of the R- and L-RWV peptides. The strain SA719, which is susceptible to most of the peptides, was used for time-dependent killing. Data are representative of 2 independent trials.

**Figure 6 antibiotics-13-00954-f006:**
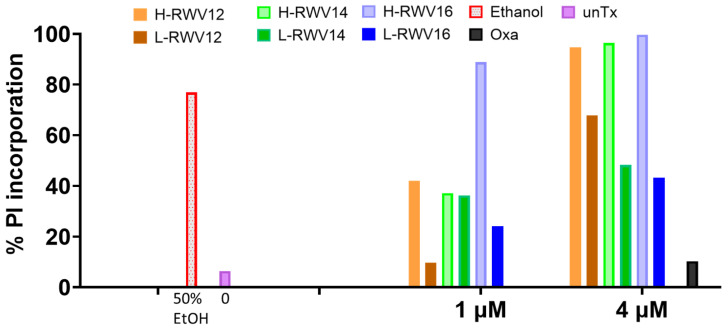
PI incorporation of RWV peptides. LRWV and H-RWV peptides from 12 to 16 residues in length were used to assess the comparative membrane-perturbing effects of the 2 series using the clinical isolate SA719. EtOH, ethanol (50%) was used as positive control and oxacillin (oxa) as negative control; unTx, untreated (0 µM); data are representative of 2 independent trials.

**Table 1 antibiotics-13-00954-t001:** Comparative primary sequences of H-RWV and L-RWV peptides (r, residues).

Charge	Length (r)	Name	H-amphipathic	Name	L-amphipathic
4	8	H-RWV8	RRWWRRWW	L-RWV8	RRRRWWWW
5	10	H-RWV10	RRWWRRVWRW	L-RWV10	RRRRRWWWWV
6	12	H-RWV12	RRVWRWVRRWWR	L-RWV12	RRRRRRWWWWVV
7	14	H-RWV14	RRVWRWVRRWWRRV	L-RWV14	RRRRRRRWWWWVVV
8	16	H-RWV16	RRVWRWVRRVWRWVRR	L-RWV16	RRRRRRRRWWWWVVVV
Blue: R, cationic residues; W and V, hydrophobic residues.					

**Table 2 antibiotics-13-00954-t002:** Mean MIC s of H-RWV and L-RWV peptides corresponding to Figure 2.

				MIC, µM			
Peptide	*S. aureus*	*E. faecium*	*A. baumannii*	*K. pneumoniae*	*Enterobacter* spp.	*E. coli*	*P. aeruginosa*
H-RWV8	29 ± 7	32	24 ± 9.2	29 ± 7.1	32	24 ± 8.8	27 ± 7.5
H-RWV10	29 ± 6.5	19 ± 13	22 ± 12	29 ± 6.5	32	22 ± 8.8	24 ± 8
H-RWV12	18 ± 12	28 ± 10	6 ± 2.3	18 ± 12	28 ± 7.4	6.4 ± 2.2	24 ± 8
H-RWV14	7.5 ± 12	17 ± 12	8 ± 5.6	7.5 ± 12	7.2 ± 5.6	2.2 ± 1.1	12 ± 4
H-RWV16	4.7 ± 5.6	2.7 ± 1	4.5 ± 2.51	4.7 ± 5.6	5.2 ± 4.7	2.2 ± 1.1	7.3 ± 1.5
L-RWV8	20. ± 0.7	30 ± 5	10.2 ± 0.9	20 ± 7.4	30.2 ± 5.3	16	30 ± 5.2
L-RWV10	26 ± 8	21 ± 9.5	8.5 ± 3.6	26 ± 8.2	32	8	22 ± 7.7
L-RWV12	3.7 ± 0.7	19 ± 11	4.6 ± 1.5	3.7 ± 0.7	6.2 ± 4	4.6 ± 1.6	14 ± 7.7
L-RWV14	3 ± 1	19 ± 11	7.4 ± 6.2	3 ± 1	16.9 ± 14	4.4 ± 2.7	21 ± 9
L-RWV16	32	32	21 ± 11	32	32	24 ± 10	32

**Table 3 antibiotics-13-00954-t003:** Minimum biofilm elimination concentrations corresponding to Figure 3.

Peptide	MBEC, µM
H-RWV12	2
H-RWV14	1.7 ± 0.5
H-RWV16	0.5 ± 0.2
L-RWV12	2.5 ± 1.3
L-RWV14	2.6 ± 1.2
L-RWV16	12 ± 5.5

## Data Availability

Data are contained within the article and Appendix A.

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
