# Peer review of "Comparative Properties of Helical and Linear Amphipathicity of Peptides Composed of Arginine, Tryptophan, and Valine"

_antibiotics, 2024, doi:10.3390/antibiotics13100954_

Round 1
Reviewer 1 Report
Comments and Suggestions for Authors
Please see the attached PDF file for detailed comments.

Experimental methods could have been more clearly written. Overall writing quality is good.
Reviewer 2 Report
Comments and Suggestions for Authors
The manuscript submitted for peer-review by Jessie Klousnitzer et al., is interesting and the reported results are convincing. The outcomes of the study are in the process of translation to product for wider therapeutic applications.
Minor concerns are:
Whether the membrane permeability and cell penetration properties of these peptides are attempted by authors.
Do these peptides have enough half-life and stability under different conditions, which would enhance the therapeutic application.
Author Response
We appreciate the reviewer's insightful comments. We are about the investigate the cyclic versions of these peptides. We will further explore the mechanisms of action in the next comparative studies with linear and cyclic derivatives. The stability issues will be also addressed in the comparative studies with cyclic derivatives.
Much appreciated.
Reviewer 3 Report
Comments and Suggestions for Authors
The authors designed and synthesized both helical and linear antimicrobial peptides (AMPs) using Arg (R), Trp (W), and Val (V) as amino acids and compared their antimicrobial activities. The manuscript is well organized. There are a few questions for the authors:
1. Lys and Phe are also commonly used amino acids in AMPs, please further explain why the reason of using RWV only.
2. In the manuscript, membrane perturbation determined only by PI incorporation measured by flow cytometry
Author Response We appreciate the reviewer's insightful comments. Lys and Phe derivatives are currently being studies in a major investigation with linear and cyclic peptides. The mechanisms of action will also be explored in depth as we anticipate potentially different mechanisms of the linear and cyclic peptides using fluorescent-labelled peptides, confocal microscopy, EM, affinity for LPS/TA/DNA using SPR, etc. This is taking much longer to investigate. Round 2
Reviewer 1 Report
Please see the attached file for detailed comments.
Reviewer 3 Report
The authors have revised the manuscript accordingly. Author Response I appreciate the reviewer's comments. A revision will be uploaded based on the latest comments. Comments.pdf
